# Vaccine-induced T cell receptor T cell therapy targeting a glioblastoma stemness antigen

Yu-Chan Chih [1,2,3,4], Amelie C. Dietsch[1,2], Philipp Koopmann[1,2], Xiujian Ma[2,5], Dennis A. Agardy [1,2,3,4], Binghao Zhao[1,2], Alice De Roia[1,2,3,4,6], Alexandros Kourtesakis[2,3,7,8], Michael Kilian [1,2,4,9], Christopher Krämer[1,2,4], Abigail K. Suwala[2,10,11], Miriam Stenzinger[12,13], Halvard Boenig[14,15], Agnieszka Blum[16], Victor Murcia Pienkowski[16], Kuralay Aman[1,2], Jonas P. Becker [2,17,18], Henrike Feldmann [1,2,3,4], Theresa Bunse [1,2,4], Richard Harbottle [2,6], Angelika B. Riemer [2,17,18], Hai-Kun Liu [2,5], Nima Etminan[19], Felix Sahm [2,10,11], Miriam Ratliff [2,8,19], Wolfgang Wick [2,7,8], Michael Platten [1,2,4,20,21,22], Edward W. Green [1,2] & Lukas Bunse [1,2,4,22] ✉

T cell receptor-engineered T cells (TCR-T) could be advantageous in glioblastoma by allowing safe and ubiquitous targeting of the glioblastoma-derived peptidome. Protein tyrosine phosphatase receptor type Z1 (PTPRZ1), is a clinically targetable glioblastoma antigen associated with glioblastoma cell stemness. Here, we identify a therapeutic HLA-A*02-restricted PTPRZ1-reactive TCR retrieved from a vaccinated glioblastoma patient. Single-cell sequencing of primary brain tumors shows *PTPRZ1* overexpression in malignant cells, especially in glioblastoma stem cells (GSCs) and astrocyte-like cells. The validated vaccine-induced TCR recognizes the endogenously processed antigen without off-target cross-reactivity. PTPRZ1-specific TCR-T (PTPRZ1-TCR-T) kill target cells antigen-specifically, and in murine experimental brain tumors, their combined intravenous and intracerebroventricular administration is efficacious. PTPRZ1-TCR-T maintain stem cell memory phenotype in vitro and in vivo and lyse all examined HLA-A*02⁺ primary glioblastoma cell lines with a preference for GSCs and astrocyte-like cells. In summary, we demonstrate the proof of principle to employ TCR-T to treat glioblastoma.

Engineered T cell therapy has constituted a great success in combating hematopoietic cancers in the last decade with more than half of the patients demonstrating clinical responses leading to long-term survival in many studies[1–3]. On the other hand, its efficacy in treating solid tumors is still limited[4]. This is largely due to poor T cell infiltration, intratumoral T cell dysfunction and limited T cell persistence. These challenges are particularly pronounced in glioblastoma, a highly malignant primary brain tumor which is compartmentalized by the blood-brain barrier (BBB) excluding even unleashed peripheral tumor-

reactive T cells, making glioblastoma one of the least responsive tumors to immunotherapies[5–8].

Glioblastoma carries relatively few mutations[9]. Given the paucity of tumor-specific cell surface antigens, only a limited number of candidate targets can be exploited. Several chimeric antigen receptor (CAR)-T cell therapy clinical trials have been launched to target glioblastoma-associated surface antigens, e.g., IL13Rα2, HER2 and GD2, with fewer focusing on the glioblastoma-specific but subclonal antigen, EGFRvIII[10,11]. Primarily driven by sophisticated synthetic

improvements of cellular approaches, some patients showed response to CAR-T cell therapy[12]. In addition, combinatorial treatments are under exploitation[13,14]. Unlike CARs, T cell receptors (TCR) enable targeting both intracellular and extracellular antigens that are processed and loaded onto the major histocompatibility complex (MHC)[15]. This nature of antigen recognition by TCRs conceivably broadens the range of targetable antigens. Moreover, in comparison to CARs, TCRs display superior antigen sensitivity, reduced tonic signaling, and engage differential proinflammatory intracellular pathways[16,17]. Encouraging results have been demonstrated in Phase I and II clinical trials using TCR-engineered T cells (TCR-T) to treat solid tumors[18]. Despite current progress in advancing cell therapy, TCR-T cell therapy has not yet been investigated in glioblastoma patients. Previously, we have isolated and validated vaccine-induced glioma-reactive TCRs targeting the tumor-associated antigen, NLGN4X[19], and neoantigens, IDH1R132H, H3K27M, and CICR215W/Q[20–22]. In a multinational European glioblastoma immunotherapy clinical trial, Glioma Actively Personalized Vaccine Consortium 101 (GAPVAC-101), newly diagnosed glioblastoma patients were immunized with a warehouse of unmutated glioblastoma-associated peptides (APVAC1), and a warehouse of individual-specific mutated peptides (APVAC2)[23]. 92% of APVAC1-vaccinated patients developed MHC class I (MHCI)-dependent immunogenicity. PTPRZ1[1814-1822], an HLA-A*02-restricted epitope derived from the glioblastoma-associated antigen protein tyrosine phosphatase receptor type Z1, PTPRZ1, was administered to four patients within APVAC1 and elicited 100% immunogenicity[23,24]. Multimer-sorted PTPRZ1[1814-1822]-T cells demonstrated cytotoxicity against HLA-A*02+ glioblastoma cell lines with PTPRZ1 expression.

PTPRZ1 is involved in central nervous system development[25,26]. Binding to its ligands, pleiotrophin (PTN) and midkine (MK), leads to oligomerization and inactivation of the phosphatase receptor, resulting in sustained phosphorylation of its substrates, such as β-catenin and anaplastic lymphoma kinase (ALK)[27,28]. Its absence leads to early onset of oligodendrocyte differentiation and disruption of the perineuronal net[25,29]. While its expression is very limited across adult tissues, it is essential for gliomagenesis. In gliomas, PTPRZ1 takes part in cancer cell proliferation, migration, and invasiveness[30–32]. In addition, it contributes to angiogenesis and tumor radioresistance[33,34]. Notably, depletion of PTPRZ1 results in significant impairment of glioma cell sphere formation in vitro and delays tumor growth in vivo, indicating a strong association of PTPRZ1 with glioma cell stemness[35,36]. Recent studies have employed PTPRZ1 along with other markers to define glioblastoma stem cells (GSC) in single-cell transcriptomic and flow cytometric analyses[37,38]. Here, we perform a comprehensive multimodal assessment of PTPRZ1 in glioblastoma, demonstrate that it is a highly attractive immunotherapeutic target, and develop an off-the-shelf TCR-T therapy which is, in principle, applicable for all HLA-A*02+ glioma patients.

## Results

### PTPRZ1 is exclusively overexpressed and presented in glioma cells, and associated with distinct cellular states and stemness in glioblastoma

To evaluate the extent of PTPRZ1 association with gliomas, we first examined its expression at transcript and protein levels in human glioblastoma tissues. In TCGA bulk RNA sequencing (RNA-seq) data, PTPRZ1 was significantly upregulated in glioblastoma compared to adjacent normal (Fig. 1a). In addition, PTPRZ1 was overexpressed in low-grade glioma (LGG) (Supplementary Fig. 1a), and its expression correlated positively with ABSOLUTE tumor purity (Supplementary Fig. 1b, c)[39]. In order to detail PTPRZ1 at cellular resolution, we interrogated publicly available glioblastoma and isocitrate dehydrogenase (IDH)-mutant glioma single-cell RNA-seq (scRNA-seq) datasets[40,41], and found PTPRZ1 to be highly expressed in malignant cells but not in immune cells such as T cells and macrophages, nor in normal glial cells,

oligodendrocytes and microglia (Fig. 1b, Supplementary Fig. 1d-h). In a paired glioma and stromal cell scRNA-seq dataset (n(glioblastoma)=16, n(IDH-mut glioma)=3, and n(pediatric high-grade glioma)=1), we independently validated malignant cell-type specificity of PTPRZ1 expression (Fig. 1c, d, Supplementary Fig. 1i, j). Interindividual levels of PTPRZ1 overexpression varied across patients but were found in all gliomas (Supplementary Fig. 1k, l). To evaluate protein abundance, we analyzed $n = 20$ matched primary and recurrent glioblastoma samples and found patient-individual PTPRZ1 levels without temporal alterations between primary and recurrent tumors (Fig. 1e, f). In concordance with transcriptomic data using GFAP to depict tumor purity, abundance of PTPRZ1+ cells correlated positively with GFAP+ cells (Fig. 1g). To address whether PTPRZ1 upregulation leads to PTPRZ1-derived epitope presentation, we performed untargeted human leukocyte antigen (HLA) ligandomics. PTPRZ1-derived ligands were found across examined primary glioblastoma cell lines, and PTPRZ1[1814-1822], in particular, was among the overlapping PTPRZ1-derived ligands (Fig. 1h, Supplementary Fig. 1i).

We recapitulated distinct cellular states in a published glioblastoma dataset (Supplementary Fig. 2a)[41], and found PTPRZ1 expression to be enriched in astrocyte-like (AC-like) and oligodendrocyte-progenitor-like (OPC-like) glioblastoma cells and to positively correlate with their module scores (Supplementary Fig. 2b, c). Similarly, in our own cohort, we found higher PTPRZ1 expression levels in AC-like and OPC-like glioblastoma cells (Fig. 1i–k). Comparable to previous results[41], the distribution of cell states differed across patients (Supplementary Fig. 2d). To assess correlation with glioblastoma stemness, a GSC score of each cell was derived from a previously defined gene set[38]. Indeed, PTPRZ1 positively correlated with GSC score in both cohorts (Fig. 1l, Supplementary Fig. 2e). Additionally, we grouped TCGA glioblastoma tumors according to dominant glioblastoma cell states (Supplementary Fig. 2f), and concluded again that PTPRZ1 positively correlated with an AC-like state and GSC (Supplementary Fig. 2g, h). Collectively, we found that PTPRZ1 expression positively correlates with AC-like and OPC-like glioblastoma states and glioblastoma stemness, and that its robust expression results in a high MHC ligand load in glioblastoma.

### A vaccine-induced T cell receptor binds intracellularly processed PTPRZ1[1814-1822] without evidence of off-target reactivity

To discover therapeutic PTPRZ1-reactive TCRs, peripheral blood mononuclear cells (PBMC) were retrieved from a female HLA-A*02+ GAPVAC-101 patient, pt.16, who had undergone 11 APVAC1 vaccinations at the time of sample collection with a favorable clinical course, showing immune responses to the examined PTPRZ1 peptides (Fig. 2a)[23]. Reactive T cells were sorted following in vitro restimulation with PTPRZ1[1347-1355] or PTPRZ1[1814-1822] peptides and then subjected to scTCR-seq. The TCR repertoire of the sorted T cells following in vitro restimulation with PTPRZ1[1347-1355] peptide was polyclonal while the one following in vitro restimulation with PTPRZ1[1814-1822] peptide was oligoclonal, with the top CDR3 taking up 75.09% and the second top 19.76% of the repertoire (Fig. 2a). Interestingly, the top CDR3 was constituted of one β chain paired with different α chains. Subsequently, the dominant TCRs for both peptides were cloned into a non-viral, non-integrating episomal scaffold matrix attachment region (S/MAR) DNA vector[42]. Constant regions of TCR α and β chains were murinized to avoid mispairing with endogenous human TCRs[43]. For subsequent TCR validation, TCRs were electroporated into Jurkat cells carrying NFAT, AP-1, and NF-κB reporters or electroporated along with a plasmid encoding a reporter into Jurkat cells (Fig. 2b, Supplementary Fig. 3a). TCR-Jurkat cells were then cocultured with peptide-loaded T2 presenter cells for 24 h. Out of the top two TCRs for PTPRZ1[1814-1822], sharing the same β chain but paired with different α chains, one demonstrated strong reactivity against peptide-loaded target cells (Fig. 2c). None of the tested dominant TCRs for PTPRZ1[1347-1355] were reactive to peptide-

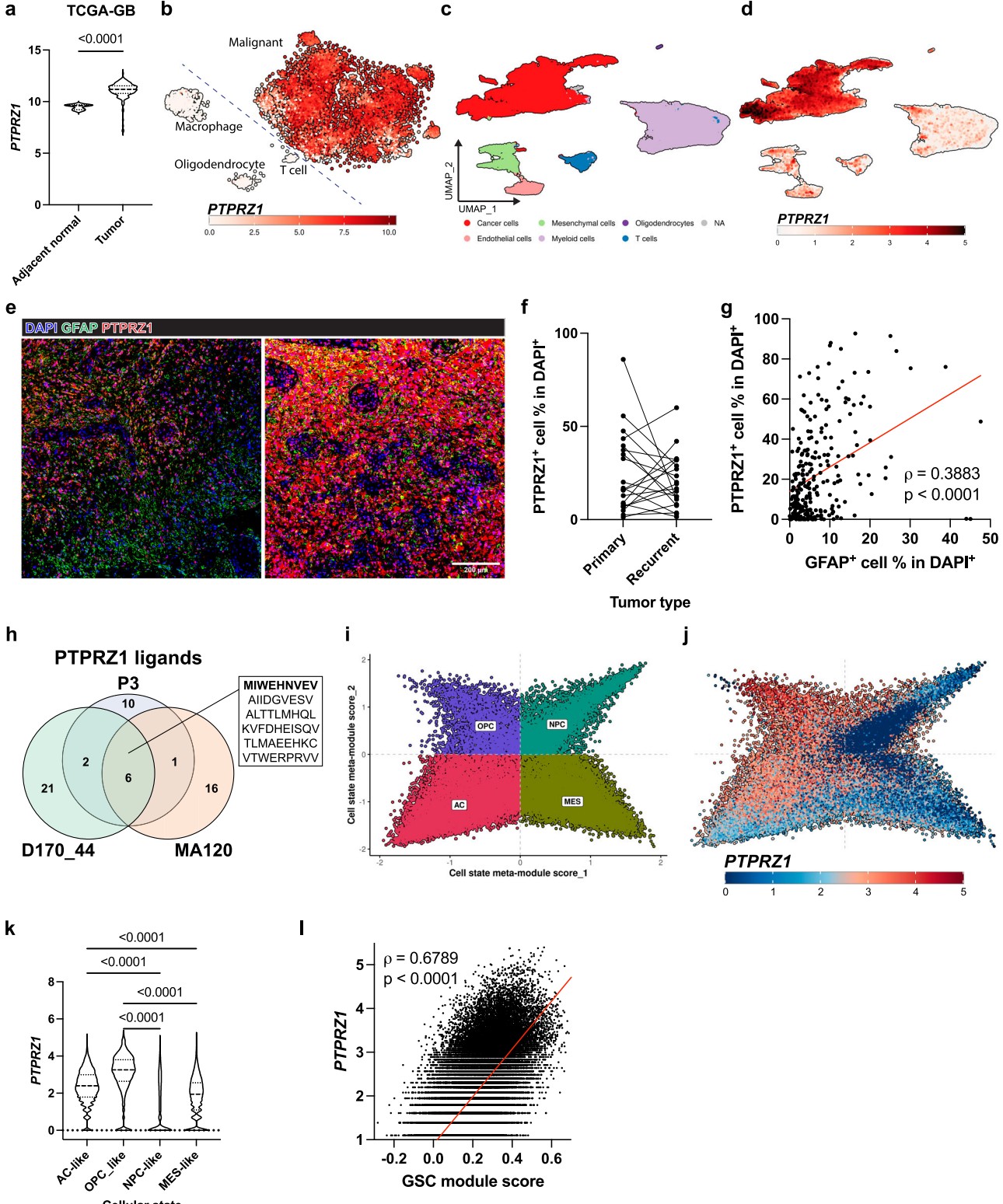

**Fig. 1 | PTPRZ1 is upregulated in glioblastoma, particularly in GSCs and AC-like glioblastoma. a** *PTPRZ1* expression in primary tumor compared to adjacent normal tissues in TCGA-glioblastoma (TCGA-GB) dataset with two-tailed t-test. **b** *PTPRZ1* expression in single cells of glioblastoma from previously published dataset[41]. Cell type annotation is shown in Supplementary Fig. 1d. **c** UMAP of scRNA-sequenced glioma and glioblastoma samples from Supplementary Fig. 1i with cell type annotation. **d** Gene expression of *PTPRZ1* for **c**. More cell type-defining genes are shown in Supplementary Fig. 1j. **e**, **f** Protein expression of PTPRZ1 and GFAP from 20 primary and 20 recurrent glioblastoma matched samples. Each dot is the average of all tumor pieces of a patient. **g** Correlation of PTPRZ1+ and GFAP+ cell frequencies. Each dot represents a tumor piece. **h** Untargeted ligandomics of primary glioblastoma cell lines with overlapping of PTPRZ1 ligands across primary glioblastoma cell lines. The highlighted overlapped peptide is PTPRZ1[1814-1822]. **i**, **j** Cellular states of cancer cells in **c** and *PTPRZ1* expression. **k**, **l** *PTPRZ1* expression across distinct cellular states and its correlation with GSC score in cancer cells from **c**. **k** was analyzed with one-way ANOVA multiple comparison corrected with Holm-Šidák method. **g** and **l** were analyzed with Spearman correlation.

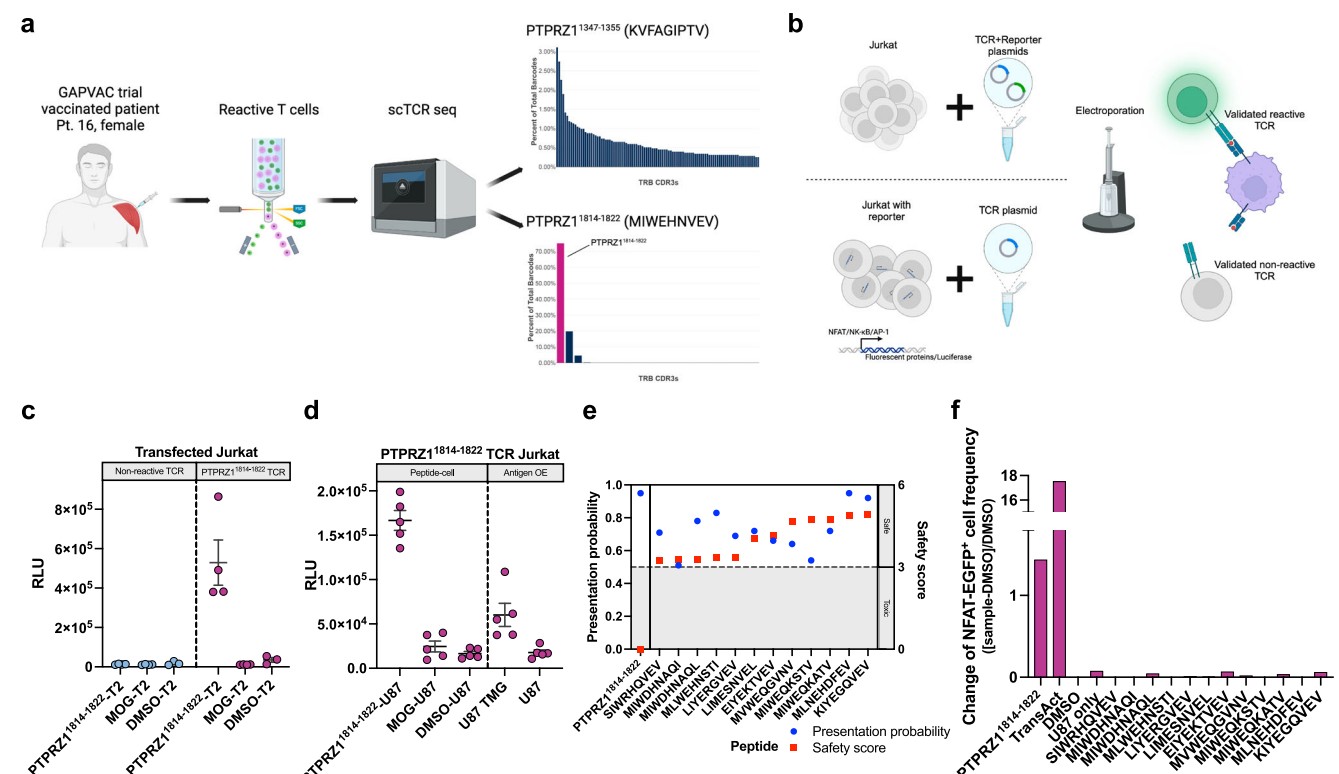

**Fig. 2 | A PTPRZ1-reactive TCR was identified from a vaccinated glioblastoma patient. a** Schematic workflow depicting the sorting and sequencing of PTPRZ1-reactive T cells with CDR3 frequency plots showing dominant TCRs. The later identified reactive TCR is highlighted. **b** Validation of dominant TCR clonotypes from **a** using Jurkat cells transfected with a TCR and a reporter plasmid (top) or Jurkat reporter cells transfected with a TCR plasmid (bottom). **c** Luminescence reporter signal of TCR-Jurkat cells upon overnight coculture with peptide-loaded T2 target cells. Each dot represents a technical replicate, n(PTPRZ1$^{1814-1822}$-T2) = 4, n(MOG-T2) = 4, and n(DMSO-T2) = 3 **d** Luminescence reporter signal following overnight coculture with peptide-loaded or antigen-expressing target cells. Each dot represents a technical replicate, $n = 5$. **e** in silico AI-predicted off-targets of the identified PTPRZ1-reactive TCR with presentation probability indicated in blue circles and safety score indicated in red squares. Presentation probability predicts whether the peptide is presented on MHC, and lower safety score denotes a higher likelihood of cross-reactivity. **f** Fluorescence reporter signal of TCR-Jurkat cells upon overnight coculture with various peptide-loaded target cells. Data are presented as mean values ± SEM. Created in BioRender. D170, P. (2025) https://BioRender.com/z55e945 (**a**); https://BioRender.com/k87g100 (**b**).

loaded target cells (Supplementary Fig. 3b, c). Thus, the identified reactive TCR for PTPRZ1$^{1814-1822}$ was subsequently further examined.

Exogenous peptides do not necessarily induce immunogenicity against naturally processed and presented antigens[44]. Therefore, we next assessed whether the peptide vaccine-induced TCR was indeed reactive to the endogenously processed and presented antigen PTPRZ1$^{1814-1822}$. Since the complete open reading frame (ORF) of *PTPRZ1* is large (~7kbp), we generated a well-characterized HLA-A*02$^+$ glioblastoma cell line (U87) stably expressing a tandem minigene (TMG) encoding several antigens derived from the GAPVAC-101 and IMA950 trials, including the *PTPRZ1* antigens of interest (Supplementary Fig. 4a, b, Supplementary Table 1)[23,45]. Upon coculture with U87 TMG cells, TCR-Jurkat cells also showed strong reporter activity (Fig. 2d), indicating that the identified TCR is able to recognize the intracellularly processed and presented antigen PTPRZ1$^{1814-1822}$. Next, we aimed to evaluate the safety profile of the PTPRZ1$^{1814-1822}$ TCR. *PTPRZ1* is, if at all, barely expressed across adult tissues[24], and the TCR was isolated from a vaccinated patient without notable adverse events[23]; hence, we hypothesized that the TCR had undergone thymic selection, making both off-target and on-target off-tumor toxicities unlikely. Nevertheless, to experimentally assess potential off-targets, we applied ARDitox, an in silico artificial intelligence (AI)-based prediction tool for off-target TCR binding[46]. A panel of potential off-targets for the PTPRZ1$^{1814-1822}$ TCR was generated with various off-target-specific safety and presentation scores (Fig. 2e, Supplementary Table 2). No high-risk (safety score < 3) potential off-targets were predicted while 12 low-risk (safety score > 3) off-targets with relevant presentation probabilities

on HLA-A*02 were predicted. In a subsequent Jurkat reporter assay, we did not find reactivity against any of these 12 potential off-targets (Fig. 2f). Together, these data reveal a patient-derived vaccine-induced PTPRZ1$^{1814-1822}$-reactive TCR that binds to both exogenous and intracellularly processed and presented antigen on MHCI and does not react against AI-predicted low-risk off-targets.

## PTPRZ1$^{1814-1822}$ TCR-engineered primary human T cells exert effector functions and maintain T$_{SCM}$ phenotype in vitro

We next endeavored to explore whether the PTPRZ1$^{1814-1822}$ TCR elicits effector functions in primary human T cells. We employed a GMP-compatible retroviral system commonly used for CAR-T cell manufacturing to transduce the constant region-murinized PTPRZ1-TCR. Assessed by flow cytometric analysis of murine constant TCR β chain surface abundance and representative for multiple experiments, transduction efficiency was > 85% for PTPRZ1-TCR and negative control Influenza (Flu)-TCR, targeting the HLA-A*02-restricted epitope GILGFVFTL (Supplementary Fig. 5a). Without further enrichment, the TCR surface expression was maintained without a decrease in frequency 20 days post transduction in vitro (Fig. 3a). Of note, our culture conditions using IL-7 and IL-15 for stimulation and maintenance instead of conventional IL-2 stimulation favored the expansion of CD8$^+$ T cells (Fig. 3b)[47]. Subsequent longitudinal subtyping of CD8$^+$ TCR-T cells into stem cell memory (T$_{SCM}$, CD45RA$^+$ CD62L$^+$), central memory (T$_{CM}$, CD45RA$^-$ CD62L$^+$), effector memory (T$_{EM}$, CD45RA$^-$ CD62L$^-$), and CD45RA$^+$ effector memory (T$_{EMRA}$, CD45RA$^+$ CD62L$^-$) revealed a long-lasting and predominant adaptation to a T$_{SCM}$ phenotype over the

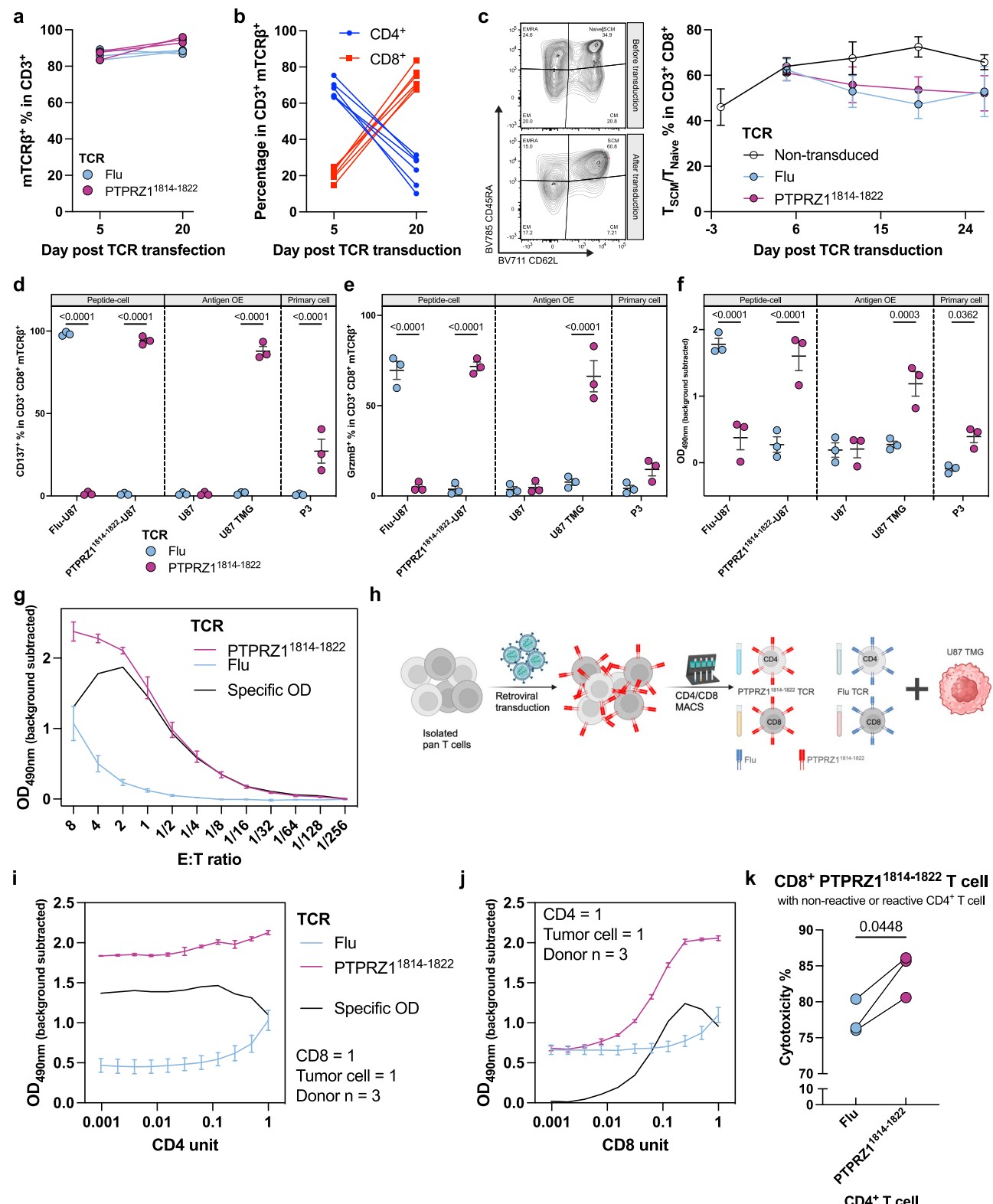

course of three weeks in vitro regardless of TCR expression (Fig. 3c). In contrast to previous studies, we did not enrich for naïve T cells yet still were able to generate $T_{SCM}$-abundant engineered T cell products[47]. The generated TCR-T cells were then cocultured with various target cells. Only upon contact with cognate peptide-loaded target cells or U87 TMG, CD8[+] TCR-T cells were activated (Fig. 3d). Importantly, the primary HLA-A*02[+] glioblastoma cell line P3, endogenously expressing *PTPRZ1*[23], also activated CD8[+] PTPRZ1[1814-1822] TCR-T cells (Fig. 3d).

Moreover, the frequencies of effector cytokine- and cytolytic protein-expressing CD8[+] TCR-T cells were increased in an antigen-specific fashion (Fig. 3e, Supplementary Fig. 5b–d). PTPRZ1[1814-1822] TCR-T cells (PTPRZ1-TCR-T) demonstrated antigen-specific dose-dependent cytotoxicity with an optimal E:T ratio of 2:1 (Fig. 3f, g).

A recent study suggests that cytotoxic CD4[+] CAR-T cells facilitate long-term tumor control[48]. Our TCR-T cell manufacturing process transduced both CD4[+] and CD8[+] T cells, and we observed that CD4[+]

**Fig. 3 | PTPRZ1^{1814-1822} TCR-T product was persistent, T_{SCM}-abundant, and efficient in target cell killing. a** TCR expression on primary human T cells over 3-week period. **b** Percentage of CD4^+ and CD8^+ T cells in TCR-T cell products over 3 weeks. **c** CD8^+ T cell subsets on the day of PBMC isolation and after transduction with longitudinal monitoring on the right. N(biological)=4. **d** Percentage of surface activation marker-positive, CD137^+, cells in CD8^+ TCR-T cells upon 24 h coculture with target cells. **e** Percentage of effector protein-positive, Granzyme B^+, cells in CD8^+ TCR-T cells upon 24 h coculture with target cells. **f** Detection of LDH released into the medium after 24 h coculture of TCR-T cells and target cells. **g** Titration of E:T ratio with TCR-T cells and U87 TMG target cells, measured with LDH release. 1 unit is 75 ×10^3 cells. Specific OD is OD_{PTPRZ1}-OD_{Flu}. **d–g** were performed with

biological replicates $N = 3$. **h** Schematic of isolating CD4^+ and CD8^+ TCR-T cells with MACS and subjecting them to various cocultures. **i** Cytotoxicity upon serial diluting CD4^+ T cells. **j** Cytotoxicity upon serial diluting CD8^+ T cells. **i** and **j** were measured with LDH release, and 1 unit equals to 75 ×10^3 cells in. **k** Cytotoxicity of coculture using reactive CD8^+ T cells with the target cell and either reactive or non-reactive CD4^+ T cells, measured with cell counting through flow cytometry. **d–f** were analyzed with two-way ANOVA multiple comparison corrected with Holm–Šidák method. **k** was analyzed with two-tailed paired t-test. Data are presented as mean values ± SEM. Created in BioRender. D170, P. (2025) https://BioRender.com/p23d083 (**h**).

TCR-T cells, even without the co-receptor CD8, were activated in an antigen-dependent manner (Supplementary Fig. 6a–c). Moreover, additional irradiation of tumor cells in vitro resulted in increased lysis of target cells likely due to enhanced activation of PTPRZ1-TCR-CD4^+-T but not PTPRZ1-TCR-CD8^+-T (Fig. 3d, e, Supplementary Fig. 5e–h and Supplementary Fig. 6c–f). Although cytolytic proteins were elevated in PTPRZ1-TCR-CD4^+-T, it remained unclear if PTPRZ1-TCR-CD4^+-T encountered target cells directly to execute cytotoxicity or to support neighboring cytotoxic PTPRZ1-TCR-CD8^+-T. Thus, we further unraveled the role of CD4^+ T cells engineered with the CD8-restricted PTPRZ1^{1814-1822} TCR on tumor cell lysis in vitro through enriching CD4^+ and CD8^+ T cells via magnetic-activated cell sorting (MACS) post transduction (Fig. 3h, Supplementary Fig. 6g, h). As expected, serial dilution of PTPRZ1-TCR-CD4^+-T in comparison to that of PTPRZ1-TCR-CD8^+-T revealed that CD4^+ T cells play a minor role in target cell killing (Fig. 3i, j). However, we found a moderate increase in cytotoxicity when both CD4^+ and CD8^+ T cells carried the PTPRZ1^{1814-1822} TCR compared to the combination of PTPRZ1^{1814-1822}-reactive CD8^+ T cells with non-target-specific CD4^+ T cells (Fig. 3k). Thus, our data suggest a differential but synergistic role for CD4^+ and CD8^+ T cells following antigen-specific activation via PTPRZ1^{1814-1822} TCR.

## PTPRZ1-TCR-T is efficacious in experimental flank and brain tumors

To investigate the therapeutic potential of the PTPRZ1^{1814-1822} TCR in vivo, we first inoculated U87 TMG cells subcutaneously (s.c.) into the flank of immunodeficient mice, followed by two doses of intravenous (i.v.) TCR-T cell administration (Fig. 4a). Following adoptive cell transfer (ACT) with PTPRZ1-TCR-T, flank tumors regressed over time while control mice without treatment or treated with an irrelevant TCR-T product showed sustained tumor growth and met termination criteria by day 40 post tumor inoculation (Fig. 4b, Supplementary Fig. 7a–d). Of note, after initial tumor regression, some PTPRZ1-TCR-T-treated animals experienced tumor recurrence starting from day 42 onwards, yet by the predefined experimental endpoint, 33% (3 out of 9) of PTPRZ1-TCR-T cell-treated animals remained tumor-free, resulting in prolonged survival (Supplementary Fig. 7c, d).

Subsequently, we advanced to investigate PTPRZ1-TCR-T in experimental brain tumors. In previous studies, differential therapeutic efficacy of cell therapy was observed in brain tumors in dependence of the route of administration[49,50]. A recent study has demonstrated that repeated systemic i.v. CAR-T delivery does not result in glioblastoma control in a phase I trial, not even in a combinatorial treatment with immune checkpoint blockade[14]. At least in part, this might be attributed to the BBB resulting in the exclusion of T cells infiltrating via the blood stream into the CNS[5,6]. Indeed, when we delivered PTPRZ1-TCR-T into mice bearing intracranial U87 TMG tumors i.v. (Supplementary Fig. 8a), no apparent therapeutic effect was observed (Supplementary Fig. 8b). Hence, we adapted our ACT regime to two doses of intracerebroventricular (i.cv.) ACT following one dose of i.v. ACT (Fig. 4c). Notably, by combined i.v. and i.cv. PTPRZ1-TCR-T, mice showed preclinical response (Fig. 4d, e). 5 out of 7 (71.4%) of PTPRZ1-TCR-T-treated tumor-bearing mice responded radiographically to ACT while tumors

of control-treated mice continued to grow (Fig. 4e). Only PTPRZ1-TCR-T-treated mice (4 out of 7; 57.1%) survived till the experimental endpoint (Fig. 4f, Supplementary Fig. 9a–c). One out of the survivors remained tumor-free macroscopically while the others experienced tumor recurrence at experimental endpoint (Supplementary Fig. 9c). To assess persistence of the TCR-T cell in vivo, we monitored T cell engraftment in the blood. Three weeks post first ACT, TCR-T cells were detected in all mice receiving ACT (Fig. 4g). Adoptive transfer of different CD8^+ T cell subsets is known to affect T cell engraftment and T cell effector functions. Paradoxically, ACT with more stem-like or central memory T cells outperforms other CD8^+ subsets as they retain self-renewal capacity but nonetheless, have less cytotoxic and cytokine-releasing capacity[51–53]. Hence, we characterized engrafted T cells by flow cytometry. Up to three weeks following first ACT, we found the T_{SCM} phenotype to remain dominant in vivo with negligible inter-individual differences (Fig. 4h, i). No difference in population frequencies was observed when comparing therapeutic PTPRZ1-TCR-T- with control TCR-T-treated mice (Fig. 4i), suggesting that the engraftment and CD8^+ T cell subset maintenance were independent of cognate antigen encounter in vivo.

To decipher if in recurrent tumors, either dormant glioblastoma cells regain proliferative capacity when reactive TCR-T cells are no longer present within the glioblastoma microenvironment[54,55], or MHC and/or antigen loss occurrs[56,57], we analyzed post-mortem tumors reaching preclinical termination criteria or experimental endpoint. We found that transferred human T cells in the tumor could only be detected in mice treated with PTPRZ1-TCR-T (Fig. 4j, k), even after 76 days following the last ACT (Supplementary Fig. 9d). In addition, MHCI expression was maintained in recurrent tumors (Fig. 4j). We thereby examined antigen expression with RNAscope™ and found that TMG transcripts were greatly diminished in the recurrent tumors (Fig. 4l, m). These data demonstrate the on-target activity of PTPRZ1-TCR-T and highlight the ACT route-dependent efficacy in experimental glioblastoma. Intriguingly, the reactive TCR-T cell product persists in experimental brain tumors, and the relapse is driven by downregulation and/or loss of the tandem antigens in this model.

## PTPRZ1-TCR-T shows cytotoxicity against patient-derived HLA-A*02^+ glioblastoma spheroid cell lines and preferentially targets slow-cycling cells

To assess PTPRZ1-TCR-T efficacy on primary glioblastoma, we established patient-derived primary glioblastoma cell lines following previously described protocols via FACS or MACS isolation (Supplementary Fig. 1i)[58]. The established primary glioblastoma cell lines were maintained in serum-free medium and cultured in spheroids to maintain stemness. Benchmarking cytotoxicity, we cocultured PTPRZ1-TCR-T with primary HLA-A*02^+ glioblastoma cell lines, D170_44 and P3, and found comparable cytotoxicities (Fig. 5a). Again, both CD8^+ and CD4^+ PTPRZ1-TCR-T cells were activated by the primary glioblastoma cell line D170_44 (Supplementary Fig. 10a, b).

Slow-cycling glioblastoma cells are considered stem-like as they have the potential for tumor initiation, therapy resistance, and generation of large number of progenies[37,59,60]. As PTPRZ1 was defined as a

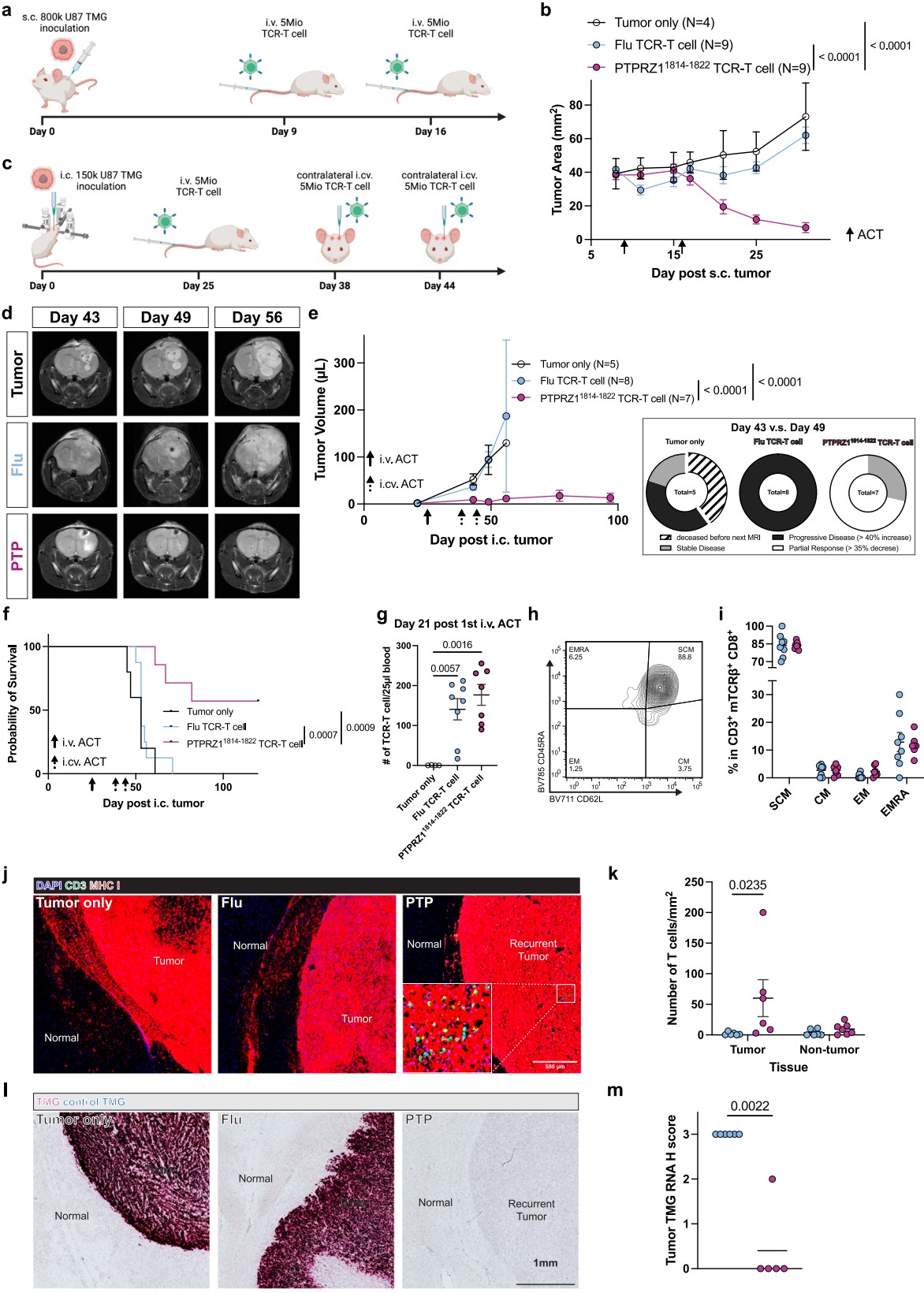

GSC marker and found to be highly associated with GSC scores, we hypothesized that PTPRZ1-TCR-T preferentially targeted stem-like slow-cycling cells (SCC). To identify SCCs, primary glioblastoma cell lines were labeled with fluorescent dye, and the top 10% dye-retaining cells after expansion were considered SCCs while the rest were defined as fast-cycling cells (FCC). Upon coculture with PTPRZ1[1814-1822] TCR-T

cells, dye-retaining SCCs and dye-losing FCCs were enumerated by flow cytometry to assess differential cytotoxicity (Fig. 5b). Indeed, in the first five hours, PTPRZ1[1814-1822] TCR-T cells preferentially killed SCCs while after 24 h, both SCCs and FCCs were lysed (Fig. 5c, Supplementary Fig. 10c), suggesting a preferential anti-tumor activity on stem-like SCCs by PTPRZ1-TCR-T.

**Fig. 4 | Intravenous and intracerebroventricular delivery of PTPRZ1-TCR-T is efficacious in experimental flank and brain tumors. a** Workflow of i.v. ACT on s.c. tumor model. **b** Longitudinal s.c. tumor growth monitoring. **c** Workflow of i.v. and i.cv. ACT on i.c. tumor model. **d** i.c. tumor imaging with preclinical MRI. **e** Longitudinal monitoring of i.c. tumor size with MRI and assessment of radiographic response upon ACT treatment. **f** Overall survival of i.c. tumor-bearing mice treated with i.v. and i.cv. ACT. **g** T cell engraftment validation with cheek blood after ACT, measured with cell counting through flow cytometry. N(Tumor only)=4, N(Flu TCR-T cell)=8, and N(PTPRZ1$^{1814-1822}$ TCR-T cell)=7. **h** Exemplary contour plot of CD8$^+$ TCR-T cell subsets of transferred T cells in host blood. **i** Percentages of different CD8$^+$ T cell subsets of transferred T cells in host blood. Same numbers of replicates were used from **g**. **j** Immunofluorescent staining of grafted i.c. tumor cells, MHC I$^+$, and transferred T cells, CD3$^+$ MHC I$^+$. **k** Transferred T cell numbers in

tumoral and non-tumoral tissues in **j** upon ACT treatment. For Flu TCR-T cell group, $N = 6$ in both tumor and non-tumor tissues. For PTPRZ1$^{1814-1822}$ TCR-T cell group, $N = 6$ in the tumor tissue, and $N = 7$ in the non-tumor tissue. **l** RNAscope™ identifying TMG expression in i.c. tumors. **m** H score analysis of TMG expression in **l**. N(Flu TCR-T cell)=6 and N(PTPRZ1$^{1814-1822}$ TCR-T cell)=5. **b** and **e** were analyzed with nonlinear regression, exponential growth equation to conclude if one curve fits all compared curves. **f** was analyzed with Log-rank test. **g** and **k** were analyzed with one-way and two-way ANOVA multiple comparison, respectively, corrected with Holm-Šidák method. **m** was analyzed with nonparametric t-test. All replicates here are biological. Data are presented as mean values ± SEM. Created in BioRender. D170, P. (2025) https://BioRender.com/b68u538 (**a**); https://BioRender.com/o44l987 (**c**).

Furthermore, to demonstrate HLA-A*02-dependency and assess the breadth of anti-tumor activity by PTPRZ1$^{1814-1822}$ TCR, we established five additional primary glioblastoma cell lines, of which MA120, MA140, and D170_108 were HLA-A*02$^+$, and MA108 and MA118 were HLA-A*02$^-$ (Fig. 5d). Upon coculture, PTPRZ1-TCR-T exclusively killed HLA-A*02$^+$ lines (Fig. 5e). Primary HLA-A*02$^+$ glioblastoma cell lines led to activation of CD8$^+$ PTPRZ1-TCR-T at various degrees and additionally activated CD4$^+$ PTPRZ1-TCR-T in a moderate fashion. Conversely, HLA-A*02$^-$ glioblastoma cell lines did not elicit any activation nor were lysed (Fig. 5e–h, Supplementary Fig. 10d). To evaluate PTPRZ1 antigen levels required for TCR-T efficacy, we next assessed the expression levels of *PTPRZ1* in our panel of generated primary spheroid glioblastoma cell lines. Importantly, in comparison to MA140, MA120 showed only moderate upregulation of *PTPRZ1* (all normalized to the well-studied cell line P3) (Supplementary Fig. 10e), but the killing was comparable (Fig. 5e). This finding suggests that PTPRZ1-TCR-T is able to lyse cells across different target expression levels. To validate target-dependent killing even at relatively low *PTPRZ1* expression levels, we generated oligoclonal *PTPRZ1* KO lines from D170_44 with two different guide RNAs (Fig. 5i). In these engineered primary glioblastoma cells, we found that a reduction of mean *PTPRZ1* expression of approximately 60% abolished TCR-T cytotoxicity (Fig. 5j). Together, these data demonstrate that PTPRZ1$^{1814-1822}$ TCR can broadly and specifically target HLA-A*02$^+$ glioblastoma primary cell lines (5 out of 5) with a preference for stem-like SCCs.

### PTPRZ1-TCR-T impacts glioblastoma cell states and targets glioblastoma stemness in individual patient tumor organoids (IPTOs)

As previously shown, *PTPRZ1* is higher expressed in AC-like and OPC-like cells and associated with glioblastoma stemness (Fig. 1k, l). Therefore, we evaluated whether PTPRZ1-enriched subsets of malignant cells were preferentially targeted by PTPRZ1-TCR-T. Individual patient tumor organoids (IPTO) maintain tumor multicellular characteristics and are thereby suitable to study PTPRZ1-TCR-T-mediated alterations in a realistic multicellular-orchestrated microenvironment (Fig. 6a). IPTOs from three HLA-A*02$^+$ patients were treated with TCR-T and interrogated by scRNA-seq. Glioblastoma cells and immune cells were identified based on the absence of EGFP and presence of B2M, respectively (Fig. 6b, Supplementary Fig. 11a, b), as host feeder organoid cells are engineered to be EGFP-expressing and B2M-deficient. The glioblastoma cell cluster expressed high levels of *PTPRZ1* (Fig. 6b). Upon PTPRZ1-TCR-T treatment, malignant cell frequency was decreased (Fig. 6c), and average *PTPRZ1* expression in malignant cells was lowered (Fig. 6d). To understand the underlying mechanism of lowered *PTPRZ1* expression, we aimed to decipher cell state distribution in post-treatment IPTOs. Assessing cell states, we again observed that *PTPRZ1* was associated with AC-like tumor cells (Fig. 6e), but upon PTPRZ1-TCR-T cell treatment, AC-like score and AC-like tumor cell frequency were reduced significantly (Fig. 6f, g). Corroborating our previous findings, a positive correlation of GSC score and *PTPRZ1* was

identified (Fig. 6h), and PTPRZ1-TCR-T coculture led to a drop of GSC cell frequency (Fig. 6i). Collectively, the results confirm that *PTPRZ1* is highly expressed in malignant brain tumor cells, most abundantly in GSCs, and that AC-like cells and GSCs are primarily targeted by PTPRZ1$^{1814-1822}$ TCR-T cells.

## Discussion

In this study, we profiled the expression of PTPRZ1, a glioblastoma-associated antigen and GSC marker, across bulk and scRNA-seq datasets in publicly available and own cohorts and took advantage of a previous multinational European glioblastoma vaccination trial, enabling us to retrieve an HLA-A*02-restricted PTPRZ1-reactive TCR in a reverse translation research paradigm. The TCR was comprehensively examined in vitro and in vivo for its anti-tumor activity against experimental flank and brain tumors, primary glioblastoma cell lines, and IPTOs. Besides the validated TCR reactivity, we identified and excluded low-risk off-targets. In addition to the robust in vivo efficacy of PTPRZ1-TCR-T, we found universal targeting of HLA-A*02$^+$ primary glioblastoma cell lines and malignant cells in IPTOs with a preference for stem-like SCCs and AC-like tumor cells.

PTPRZ1 is essential during neurodevelopment but not in adults[25,26]; however, its expression re-emerges and is required for gliomagenesis and tumor progression. Whereas the mechanistic relevance of PTPRZ1 in glioblastoma remains elusive, its functions and contributions to disease malignancy are well-demonstrated, ranging from cancer cell proliferation and stemness to angiogenesis and therapy resistance[30–34]. Here, we explored treatment with the identified PTPRZ1$^{1814-1822}$ TCR, yet therapies for non-HLA-A*02$^+$ patients and combinatorial therapies should be investigated further, particularly when taking the variety of PTPRZ1-derived MHCI-ligands into consideration (Fig. 1h). Apart from PTPRZ1 upregulation in glioblastoma, we also found robust overexpression of PTPRZ1 in IDH-mutant gliomas. Further investigations are required to examine whether PTPRZ1-TCR-T can be applied to oligodendrogliomas and astrocytomas or other cancer entities that display elevated *PTPRZ1* expression[61]. Moreover, our results further support PTPRZ1 as a GSC marker, not only from the bioinformatic analyses but also from in vitro assays showing preferential killing of stem-like SCCs. In addition, a previous study revealed that tumor-associated macrophages secrete PTN, which is tumor-promoting and positively associated with *PTPRZ1* expression and stemness of glioblastoma cells[36]. From a neuro-oncological perspective, targeting a cell population that drives disease initiation, progression, and treatment resistance, supports future clinical investigations of PTPRZ1-TCR-T.

A major advantage of TCR-T therapy is its capability to target all proteins, including intracellular neoantigens, as long as they are processed and presented on MHC. As studied by others, PTPRZ1, a transmembrane receptor, can be, in principle, targeted by CAR-T cells[62]. However, protein glycosylation is often altered in cancers, including brain tumors[63,64]. Such aberrant post-translational modification may impact CAR-T cell therapy efficacy[65], and increasing evidence is

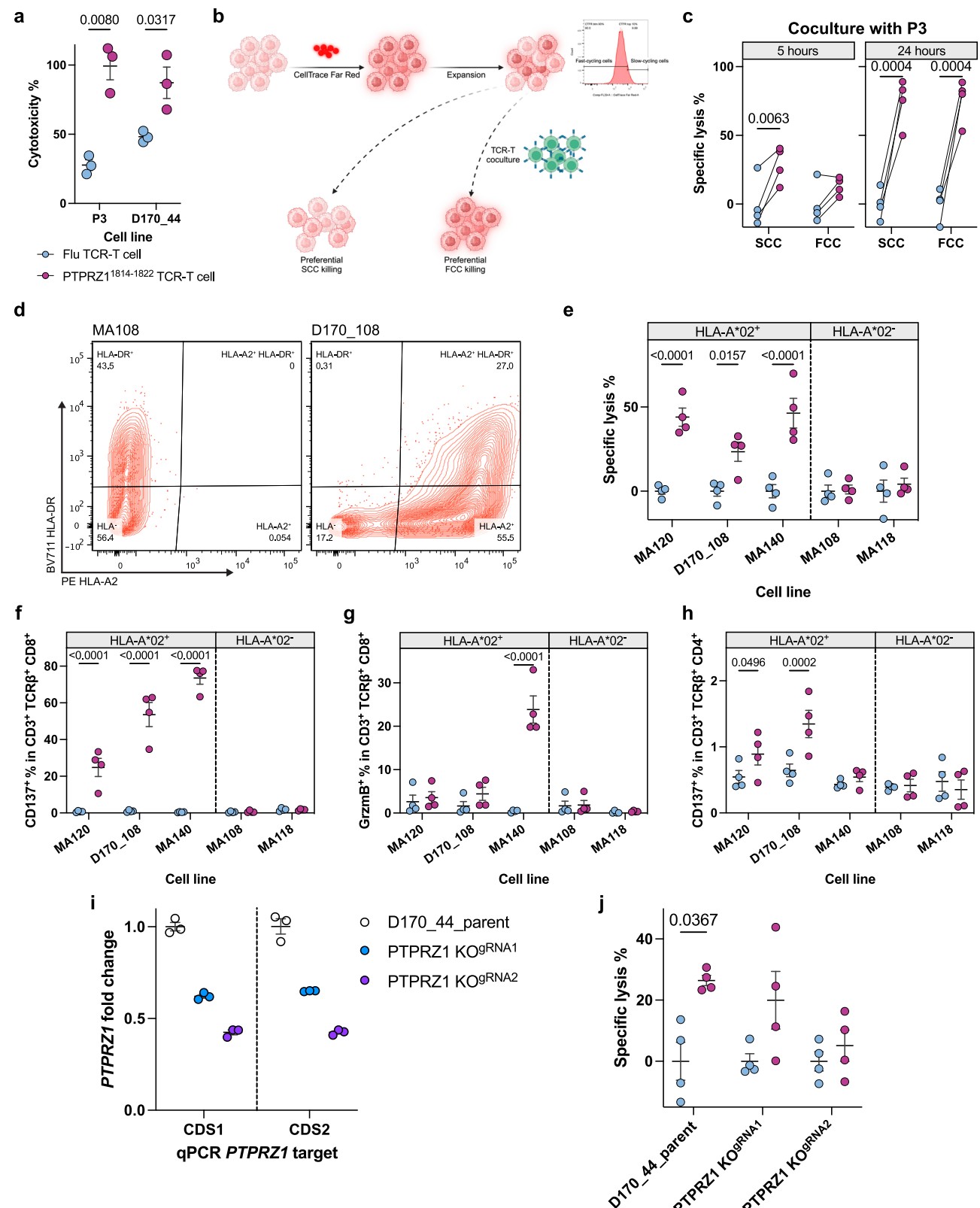

emerging suggesting TCR but not tonic CAR signaling cascades to be associated with favorable intratumoral T cell fate[16,17]. Conversely, limitations of TCR-T are HLA-type dependency and the prerequisite of MHC expression. Despite HLA restriction, which is inherently associated with the need of therapy individualization, our TCR covers a common HLA and targets the intracellular domain of PTPRZ1 presented on MHCI. As MHC antigen processing and presentation machinery is rarely altered in glioblastoma[57], TCR-T cell therapy presents a highly promising approach. Of note, in our glioma model overexpressing TMG in vivo, we found loss of the cognate antigen (Fig. 4l). It remains speculative if this is specific to our model as *PTPRZ1* expression remained stable during disease progression in patients (Fig. 1f). Nevertheless, more glioblastoma-reactive TCRs should be developed to expand the HLA coverage and to allow multivalent cell therapy, preventing escape

**Fig. 5 | PTPRZ1[1814-1822] TCR-T cells broadly lyse HLA-A*02+ primary glioblastoma cells, particularly stem-like SCCs. a** Cytotoxicity measured with LDH release of primary cell line upon coculture with TCR-T cells. N(biological)=3. **b** Experimental design to assess preferential killing of TCR-T cells on dye-retaining SCCs or dye-losing FCCs. **c** Assessment of specific lysis of SCCs and FCCs upon short-term, 5 h, or long-term, 24 h, coculture with TCR-T cells, measured with cell counting by flow cytometry. **d** HLA-A2 typing of established glioblastoma primary cell lines. **e** Assessment of specific lysis by PTPRZ1[1814-1822] TCR-T cells normalized to Flu TCR-T cell-treated HLA-A*02+ or HLA-A*02- tumors. **f** and **g** Activation of CD8+ TCR-T cells after 24 h coculture with various glioblastoma primary cell lines that are HLA-A*02+ or HLA-A*02-. **h** Activation of CD4+ TCR-T cells after 24 h coculture with

glioblastoma primary cell lines. **i** *PTPRZ1* expression levels in parental and CRISPR *PTPRZ1* KO D170_44 cell lines measured with RT-qPCR. The expression levels were normalized to the parental line, and two qPCR targets were assessed for two different coding sequences (CDS) of *PTPRZ1*. n(technical)=3. **j** Assessment of specific lysis of cell lines as in **i** by PTPRZ1[1814-1822] TCR-T cells normalized to Flu TCR-T cell-treated samples. In **c**, **e–h**, and **j**, N(biological)=4. All analyses were performed with two-way ANOVA multiple comparison corrected with Holm-Šidák method. Blue, Flu TCR-T control cells; purple, PTPRZ1[1814-1822] TCR-T cells in **a**, **c**, **e–h**, **j**). Data are presented as mean values ± SEM. Created in BioRender. D170, P. (2025) https://BioRender.com/z72g103 (**b**).

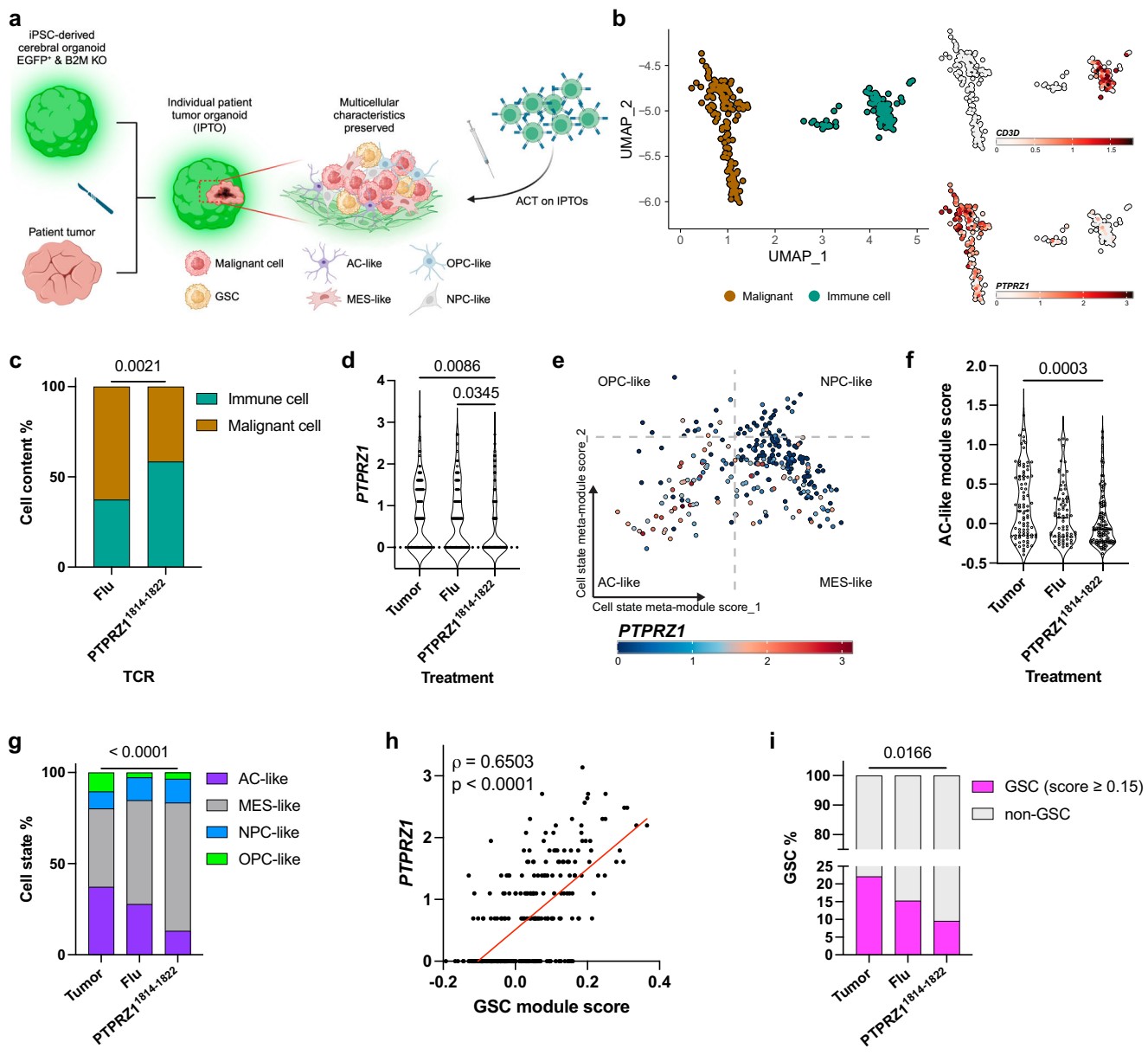

**Fig. 6 | IPTOs reveal predominant targeting of AC-like cells and GSCs by PTPRZ1[1814-1822] TCR-T cells. a** IPTO generation from three HLA-A*02+ glioblastoma samples and treatment with TCR-T cells. **b** UMAP of malignant cells and immune cells derived from Supplementary Fig. 11a with their marker expression on the right. More canonical markers are visualized in Supplementary Fig. 11b. **c** Frequency of cell content in IPTOs upon TCR-T cell treatment. **d** *PTPRZ1* expression after TCR-T cell treatment. **e** Cellular states of generated IPTOs with *PTPRZ1* expression. **f** and

**g** AC-like module score and frequency of distinct cellular states after ACT. **h** and **i** Correlation of *PTPRZ1* expression and GSC module score in malignant cells in IPTOs and frequency of GSC upon TCR-T cell treatment. Cells score over 0.15 GSC module score are defined as GSCs. **d** and **f** were analyzed with one-way ANOVA multiple comparison corrected with Holm-Šidák method. **c**, **g** and **i** were analyzed with two-sided Fisher's exact test. **h** was analyzed with spearman correlation. Created in BioRender. D170, P. (2025) https://BioRender.com/k21a580 (**a**).

through antigen loss. In this work, constant regions of the TCR were murinized to avoid potential mispairing with endogenous TCRs. Whereas a clinical study has described humoral responses against murine TCR variable regions in a subset of patients[66], the generation of these antibodies was not associated with persistence of the T cell product nor therapy response. Nevertheless, both humoral and cellular immune responses against murinized TCRs should be evaluated in future clinical trials applying this technology.

In our study, the presence of CD4+ T cells within PTPRZ1-TCR-T was required for optimal cytotoxicity in vitro. In line with this, the relevance of the CD4+::CD8+::APC (antigen presenting cell) immune triad for solid tumor eradication has been highlighted recently[67]. In PTPRZ1-TCR-T, CD4+ T cells also produced effector proteins and cytokines, suggesting a dual action of moderate direct cytotoxicity and paracrine reprogramming of CD8+ T cells. Whether CD4+ T cells engineered with CD8-restricted TCR can hijack tumor cells to form such a triad to restore and improve CD8+ T cell functions will require further experimental investigations. Notably, CD4+ CAR-T cell persistence was associated with long-term survival in leukemia patients, but on the other hand, CD4+ CAR-T cells are implicated in cytokine release syndrome. Here, fine tuning of CD4+:CD8+ T cell ratio without impacting anti-tumor immunity is required[48,68].

Encouraged by the preclinical results, we are initiating a phase I clinical trial, Intraventricular T cell receptor transgenic T cell therapy to treat glioblastoma (INVENT4GB)[69], to assess the feasibility and safety of intravenous and intracerebroventricular CD4+/CD8+ PTPRZ1-TCR-T cell therapy in patients with recurrent glioblastoma. Although i.v. delivery of TCR-T cells in our experimental glioblastoma model was not efficacious, with our current knowledge, we cannot exclude its potential efficacy in glioblastoma patients. On a global scale, it will mark the first-in-human TCR-T cell therapy against glioblastoma and highlight the here proposed "bedside to bench to bedside" approach. Prospectively, as a state-of-the-art immunotherapeutic modality which has, to our knowledge, not yet been applied to glioblastoma patients, TCR-T opens new avenues in regards to combinatorial treatment regimens including radiotherapy and immunomodulation for glioblastoma patients.

## Methods

### Cell lines

U87, purchased from ATCC[70], was cultured in DMEM supplemented with 10% fetal bovine serum (FBS) and 1% Penicillin/Streptomycin (P/S). U87 TMG cell line was generated by transfection with pMXs-IRES-PuroR plasmid encoding TMG as illustrated in Supplementary Fig. 4a. Transfection was performed with Fugene HD transfection reagent (#E2312, Promega) following the manufacture instructions. Briefly, U87 cells were seeded at a density of $3 \times 10^5$ cells per well in a 6-well plate and rested for 24 h. On the next day, the medium was replenished, and the cells were transfected with 2 μg of the DNA plasmid and rested for another 48 h. Cells stably expressing TMG were then selected with 2 μg/ml Puromycin. T2 cells were kindly provided by Dr. Angelika Riemer, Division of Immunotherapy and Prevention, DKFZ, and cultured in RPMI-1640 supplemented with 20% FBS and 2% L-Glutamine.

HEK293 cells were cultured in IMDM supplemented with 10% FBS.

Jurkat TCR deficient cells were purchased from ATCC, and Jurkat 76-Triple parameter reporter (J76-TPR) was kindly provided by Prof. Peter Steinberger, Division for Immune Receptors and T Cell Activation, Institute of Immunology, Medical University of Vienna[71]. Jurkat cells were cultured in RPMI-1640 supplemented with 10% FBS and 1% P/S.

Primary brain tumor samples were collected at the University hospital of Mannheim. All patients have provided written signed informed consent in accordance to the WMA Declaration of Helsinki principles. Ethical approval for the isolation of brain tumor tissue and analyses was obtained from the Mannheim Medical Faculty Ethics Committee (2017-589N-MA, 608-22, 574-23). The harvested tumors

were processed with tumor dissociation kit (#130-095-929, Miltenyi Biotec) and further enriched for tumor cells with tumor cell isolation kit (#130-108-339, Miltenyi Biotec) following the manufacture instructions. The isolated tumor cells and previously established primary glioblastoma cell lines were cultured in DMEM-F12 supplemented with B27 (#17504044, Thermo Fisher), 5 μg/ml Insulin (#I9278, Sigma-Aldrich), 5 μg/ml Heparin (#H4784, Sigma-Aldrich), 20 ng/ml epidermal growth factor (EGF, #AF-100-15, Peprotech) and 20 ng/ml fibroblast growth factor (FGF, #100-18b, Peprotech)[58].

To generate knockout cell lines from D170_44, gRNA sequences targeting PTPRZ1 (ATGGTATCATAAACGACTCGAGG and GAAGGCGCTATTGTGAATCCTGG) were designed using the CHOP CHOP online CRISPR design tool[72]. Top two gRNAs without self-complementarity were selected and cloned into the lentiCRISPRv2-blast construct (Addgene #83480; RRID:Addgene 83480). Lentivirus was produced with HEK293 cells which were transfected with $4.7 \times 10^{11}$ molecules of the cloned lentiCRISPRv2blast construct along with psPAX2 (Addgene #12260;RRID:Addgene 12260) and VSV-G envelope-expressing plasmid pMD2.G (Addgene #12259;RRID:Addgene 12259) using Fugene HD transfection reagent. The supernatant was collected 24 h and 48 h post-transfection, filtered, and concentrated with PEG-it Virus Precipitation Solution (#LV810A-1, Systems Bioscience). D170_44 cells were then transduced in the presence of 8 μg/ml Polybrene (#TR-1003-G, Sigma-Aldrich), and cells were selected and maintained with 1 μg/ml Blasticidin (#A1113903, Gibco).

All cell lines were incubated at 37 °C and 5% CO$_2$ with saturated humidity.

### Mice

NOD-$Prkdc^{scid}$-$Il2rg^{Tm1}$/Rj (NXG) mice were purchased from Janvier labs. NOD.Cg-$Prkdc^{scid}$ $H2$-$K1^{tm1Bpe}$ $H2$-$Ab1^{1em1Muw}$ $H2$-$D1^{tm1Bpe}$ $Il2rg^{tm1Wjl}$/SzJ (NSG MHC KO) mice were obtained from LD Schultz[73], The Jackson Laboratory, and bred at the DKFZ animal facility. Mice were housed under Specific and Opportunistic Pathogen Free (SOPF) conditions and under 12 h day/night cycle with water and chow ad libitum. Both male or female mice were used due to breeding and availability, at a minimum age of 7 weeks and a maximum age of 30 weeks. All animal procedures were conducted in compliance with the institutional laboratory animal research guidelines and were approved by the governmental institutions (Regional Administrative Authority Karlsruhe, Germany, file numbers: G-263/18 and G-130/23).

### IPTO generation and culture

Induced pluripotent stem cells (iPSCs; AICS-0036-006, Institute for Cell Science) expressing enhanced green fluorescent protein (EGFP) were utilized to generate cerebral organoids. Organoid generation adhered to a published method[74], beginning with the seeding of dissociated iPSCs in 96-well round bottom ultra-low attachment plates with previously described hESC medium[74], supplemented with 4 ng/ml basic fibroblast growth factor (bFGF; #PeproTech, 100-18B), and 50 μM ROCK inhibitor (#72304, Stemcell Technologies). Formed embryoid bodies were subsequently transferred to 24-well ultra-low attachment plates and 6-well ultra-low attachment plates in previously described Neural Induction Medium[74], followed by a transition to improved differentiation medium -A and improved differentiation medium +A[75], with agitation introduced from day 18. Resected tumor tissues were processed and dissected into small explants, which were subsequently inserted into incised cerebral organoids and embedded in Matrigel. Generated IPTOs were then cultured in improved differentiation medium +A on an orbital shaker in an incubator (75 rpm, 37 °C). After 10 days of incubation, the IPTOs were ready for TCR-T testing.

### Electroporation of Jurkat cells

Neon transfection system (Thermo Fisher) was employed to deliver TCR and reporters. For transfection of Jurkat cells without reporters,

$2 \times 10^6$ Jurkat cells were resuspend in 100 µl R buffer with 5 µg of TCR and 5 µg of reporter plasmids. For transfection of J76-TPR, $2 \times 10^6$ cells were resuspend in 100 µl R buffer with 5 µg of TCR plasmid. Electroporation was then performed at 1325 V, 3 pulses, 10 ms.

### Transduction of primary human T cells

Retroviral transduction was performed as previously described[76]. Briefly, TCR were inserted into SFG-IRES-GFP retroviral vector (kind provision from Dr. Martin Pule, Addgene #22493; RRID:Addgene_22493) with In-Fusion Cloning (#638947, Takara). $3 \times 10^6$ HEK293 cells were seeded a day before transfection in 10 ml IMDM supplemented with 10% FBS in 60.1 cm² dish. On the day of transfection, 3.75 µg TCR-SFG along with 3.75 µg PeqPam and 2.5 µg RD114 packaging plasmids were resuspended in 470 ml IMDM and 30 µl Fugene HD transfection reagent. PeqPam and RD114 packaging plasmids were kindly gifted by Dr. Tim Sauer from the Department of Hematology, Oncology and Rheumatology at Heidelberg University Hospital. After 10-min incubation, cells were transfected with the DNA-Fugene HD mix and incubated for 48 h. The viral supernatant was then collected and strained through 0.45 µm strainers. Human T cells were obtained from healthy donor PBMCs (German Red Cross Blutspendedienst Mannheim (608-22)) via density gradient separation followed by MACS with Pan T cell isolation kit (#130-096-535, Miltenyi) in accordance with manufacturer instructions. Isolated T cells were activated in CTL medium (45% RPMI-1640, 45% Click's medium, 10% FBS, 10 ng/ml IL-7 [#200-7, Peprotech] and 5 ng/ml IL-15 [#200-15, Peprotech]) with T Cell TransAct (#130-111-160, Miltenyi) for 48 h at a concentration of $1 \times 10^6$ cells/ml. The strained viral supernatant was plated 0.5 ml per well in a non-tissue culture treated 24-well plate (#351147, Falcon) precoated o/n with 0.5 ml of 7 µg/ml RetroNectin (#T100B, Takara) and centrifuged at $2000 \times g$ for 90 min at 4 °C. Later, the supernatant was removed and activated T cells were seeded at a concentration of $0.5 \times 10^6$ cell/ml in 1 ml CTL medium per well and centrifuged at $500 \times g$ for 5 min. After 4 days of incubation, TCR-T cells were ready for expression analysis or coculture assays. TCR-T cells were maintained in culture with CTL medium over the course of described time, and the medium was refreshed every 3-4 days.

### Subcutaneous tumor inoculation

U87 TMG cell line was prepared at a concentration of $4 \times 10^6$ cells/ml in PBS-Matrigel mixture at a ratio of 1:1. Immunodeficient mice were shaved at the flank site before injecting subcutaneously 200 µl of prepared tumor cells ($8 \times 10^5$ cells) with 27 G needle slowly and steadily into it. Tumor-bearing mice were then monitored routinely for tumor-related symptoms and measured for tumor growth with a caliper. Upon termination criteria or the experimental endpoint, mice were sacrificed with anesthesia and organs of interest were harvested for downstream analyses.

### Intracranial tumor inoculation

U87 TMG cell line was resuspended at a concentration of $75 \times 10^6$ cells/ml in PBS and 2 µl of which, namely $1.5 \times 10^5$ cells, was stereotactically implanted into the right hemisphere of immunodeficient mice with the following coordinates: 2 mm right lateral of the bregma and 1 mm anterior to the coronal suture at the depth of 3 mm below the dural surface. A 10-µl Hamilton micro-syringe driven by a fine-step stereotactic device (Stoelting) was employed for injection. The surgery was conducted under anesthesia (Ketamin, 100 mg/kg i.p. and Xylazin, 10 mg/kg i.p.) and analgesia (Carprofen, 5 mg/kg s.c.). Mice continued to receive analgesia for 3 days post surgery and were checked daily for tumor-related symptoms. Upon termination criteria or the experimental endpoint, mice were sacrificed with anesthesia and organs of interest were harvested for downstream analyses.

### Intravenous adoptive cell transfer

TCR-T cells generated as described above were resuspended at a concentration of $25 \times 10^6$ cells/ml in PBS. Mice were shortly warmed with red-light lamp before intravenously receiving ACT of $5 \times 10^6$ cells in 200 µl PBS with 27 G needle. On the day of and the day after ACT, mice were given $5 \times 10^4$ units of IL-2 i.p. in 100 µl PBS.

### Intracerebroventricular adoptive cell transfer

$5 \times 10^6$ TCR-T cells were resuspended in 4 µl PBS and stereotactically injected into the cerebral ventricle of the mice with the following coordinates: 0.5 mm left lateral to the bregma at the depth of 1.8 mm below the dural surface. A 10-µl Hamilton micro-syringe driven by a dine step stereotactic device (Stoelting) was employed for injection. The surgery was conducted under anesthesia (Ketamin, 100 mg/kg i.p. and Xylazin, 10 mg/kg i.p.) and analgesia (Carprofen, 5 mg/kg s.c.). Mice continued to receive analgesia for 3 days post surgery.

### Adoptive cell transfer on IPTO

Upon the establishment of IPTOs, $150 \times 10^3$ TCR-T cells were injected in 3 µl with a 10-µl Hamilton micro-syringe. After 3-day incubation, feeder cells were first macroscopically removed with a scalpel, and the remained tumor chunk was processed with tumor dissociation kit (#130-095-929, Miltenyi Biotec).

### Magnetic resonance imaging

MRI was conducted by the small animal imaging core facility at DKFZ with a Bruker BioSpec 3Tesla (Ettlingen, Germany) with Para Vision software 360 V1.1. Mice were anesthetized with 3.5% sevoflurane in air, and the imaging was performed with a T2 TurboRARE sequence: TE = 48 ms, TR = 3350 ms, FOV 20 × 20 mm, slice thickness 1 mm, averages = 3, Scan Time = 3m21s, echo spacing 12 ms, rare factor 8, slices 20, image size 192 × 192. Tumor volume was assessed by manual segmentation using Bruker Para Vision software 6.0.1.

### Untargeted ligandomics

Immunoprecipitation of HLA class I:peptide complexes was performed as previously described with additional steps for the forced oxidation of methionine using $H_2O_2$ and reduction and alkylation of cysteine using tris(2-carboxyethyl)phosphine and iodoacetamide[77]. Lyophilized peptides were dissolved in 12 µl of 5% ACN in 0.1% TFA and spiked with 0.5 µl of 100 fmol/µl Peptide Retention Time Calibration Mixture and 10 fmol/µl JPTRT 11, a subset of peptides from the Retention Time Standardization Kit (JPT), and transferred to QuanRecovery Vials with MaxPeak HPS (Waters, Milford, MA, USA). All samples were analyzed using an UltiMate 3000 RSLCnano system coupled to an Orbitrap Exploris 480 equipped with a FAIMS Pro Interface (Thermo Fisher). For chromatographic separation, peptides were first loaded onto a trapping cartridge (Acclaim PepMap 100 C18 µ-Precolumn, 5µm, 300 µm i.d. x 5 mm, 100 Å; Thermo Fisher) and then eluted and separated using a nanoEase M/Z Peptide BEH C18 130 A 1.7 µm, 75 µm × 200 mm (Waters). Total analysis time was 120 min, and separation was performed using a flow rate of 0.3 µl/min with a gradient starting from 1% solvent B (100% ACN, 0.1% TFA) and 99% solvent A (0.1% FA in $H_2O$) for 0.5 min. Concentration of solvent B was increased to 2.5% in 12.5 min, to 28.6% in 87 min and then to 38.7% in 1.4 min. Subsequently, concentration of solvent B was increased to 80% in 2.6 min and kept at 80% solvent B for 5 min for washing. Finally, the column was re-equilibrated at 1% solvent B for 11 min. The LC system was coupled on-line to the mass spectrometer using a Nanospray-Flex ion source (Thermo Fisher), a SimpleLink Uno liquid junction (FossilIonTech) and a CoAnn ESI Emitter (Fused Silica 20 µm ID, 365 µm OD with orifice ID 10 µm; CoAnn Technologies). The mass spectrometer was operated in positive mode and a spray voltage of 2400 V was applied for ionization with an ion transfer tube temperature of 275 °C. For ion mobility

separation, the FAIMS module was operated with standard resolution and a total carrier gas flow of 4.0 l/min. Each sample was injected twice using either a compensation voltage of −50 V or −65 V for maximal orthogonality and thus increased immunopeptidome coverage. Full Scan MS spectra were acquired for a range of 300–1650 m/z with a resolution of 120.000 (RF Lens 50%, AGC Target 300%). MS/MS spectra were acquired in data-independent mode using 44 previously determined dynamic mass windows optimized for HLA class I peptides with an overlap of 0.5 m/z. HCD collision energy was set to 28% and MS/MS spectra were recorded with a resolution of 30.000 (normalized AGC target 3000%). MS raw data was analyzed using Spectronaut software (version 17.6, Biognosys)[78], and searched against the Uni-ProtKB/Swiss-Prot database (retrieved: 21.10.2021, 20387 entries). Search parameters were set to non-specific digestion and a peptide length of 7 -15 amino acids. Carbamidomethylation of cysteine and oxidation of methionine were included as variable modifications. Results were reported with 1% FDR at the peptide level. Peptides identified by Spectronaut were further analyzed using NetMHCpan 4.1[79]. Predicted non-binders were excluded from the analysis.

## RNAscope™

Target RNA transcripts on slides were detected with RNAscope 2.5 HD Duplex Kit (#322435, Advanced Cell Diagnostics) following manufacture instructions. Briefly, slides with sectioned frozen murine brains were removed from −80 °C and immediately fixated with pre-chilled 10% neutral buffered formalin at 4 °C. To dehydrate the slides, they were then serially immersed in 50%, 70% and 100% EtOH at RT. After another immersion in fresh 100% EtOH, slides were either immediately used or stored in 100% EtOH at −20 °C for up to a week. To stain the slides, they were first air-dried and pre-treated with RNAscope Hydrogen Peroxide. After a quick wash with PBS, they were then pre-treated with RNAscope Protease IV. Slides were subsequently quickly rinsed with PBS. For target detection, custom probes were manufactured, and the probe mix was applied on slides and incubated in HybEZ Oven for 2 h at 40 °C. Next, the slides were washed with 1X Wash Buffer and kept in 5X SSC at RT o/n. On the next day, slides were washed with 1X Wash Buffer and underwent a series of amplification steps with Amp 1–6 following the manual. First probe was then detected with FastRed. After more washes with 1X Wash Buffer, slides underwent another series of amplification steps with Amp 7–10 following the instructions. Second probe was then detected with FastGreen. After more washes with 1X Wash Buffer, slides were counterstained with 50% Hematoxylin staining solution. Next, slides were immediately rinsed with tap water, followed by drying in HybEZ Oven at 60 °C. Once the slides cooled down, they were briefly dipped in fresh Xylene and mounted with VectaMount Mounting Medium (#H-5000, Vector Labs).

## ARDitox off-target prediction

Off-targets were predicted as described previously[46]. Briefly, 9-mer peptides of human proteome that share at least 5 amino acids with the target epitope were shortlisted. Next, epitopes derived from frequent single nucleotide polymorphisms with a frequency more than 1% were included. High-affinity presented epitopes predicted with ARDisplay were selected[80]. Safety score compared the physico-chemical properties of probable TCR-facing amino acids.

## TCGA and publicly available dataset analysis

TCGA data were downloaded through the package TCGAbiolinks (2.28.4)[81]. Sc glioma datasets were downloaded according to the instructions[40,41]. The data were handled with Seurat (5.0.3)[82]. Cell state scores were directly used if specified in the dataset; otherwise, they were calculated based on the defined gene sets with AddModuleScore function[41]. GSC scores were calculated with the published gene set[38].

## Single-cell RNA-seq and analysis

Isolated primary tumor cells and non-tumor cells from brain tumor samples and IPTO cells were resuspended in 0.04% BSA in PBS; Up to 20 ×10³ cells were loaded for 5' single cell sequencing (#1000263, 10x Genomics), and the libraries were prepared according to the manufacturer protocol. Single-cell data were aligned with cellranger (7.0.0) and handled with Seurat (5.0.3). Cells expressing few transcripts or genes were excluded before normalization. Doublets were identified and excluded with scDblFinder (1.14.0)[83]. Harmony (1.0.3) was subsequently used to integrate datasets[84]. Canonical markers were employed to identify and annotate the cell type. Module gene sets were derived from previous studies and their scores were calculated using AddModuleScore function in Seurat. Plots were made with ggplot2 (3.5.0) and SCpubr (2.0.2)[85,86].

## Statistical analysis and figures

Data are presented as individual values or as mean ± SEM unless stated otherwise. Applied statistical tests are indicated in Figure legends. GraphPad Prism 9.0 was used for statistical tests and plots. Some Figures were created with BioRender.com.

## Reporting summary

Further information on research design is available in the Nature Portfolio Reporting Summary linked to this article.

## Data availability

The use of the primary tumor cell lines and the single-cell transcriptomics of primary patient tumors specified in this manuscript are restricted by patient informed consent and institutional review board approval to this study. The processed data will be made available to academic researchers upon request. Single cell RNA-seq data are retrieved from https://doi.org/10.1126/science.aai8478 and https://doi.org/10.1016/j.cell.2019.06.024. Source data are provided with this paper.

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

## Acknowledgements

We acknowledge the support of the DKFZ Light Microscopy Facility, Genomics and Proteomics Core Facility, Center for Preclinical Research, Flow Cytometry Core Facility and single-cell Open Lab. We also would like to highlight and acknowledge the technical support from Viktoria Sachs, Chris Pagel, Kristine Jähne, and Rebecca Köhler. This study was supported by grants from the Swiss Cancer Foundation (Swiss Bridge Award), the Else Kröner Fresenius Foundation (2019_EKMS.49), the University Heidelberg Foundation (Hella Bühler Award), the DFG (German Research Foundation), project 404521405 (SFB1389 UNITE Glioblastoma B03), the DKFZ Hector institute (T-SIRE), the Hertie Foundation, and the University of Heidelberg, ExploreTech! Grant to L.B.; the DKTK Joint Funding AMI2GO, the Rolf Schwiete Foundation (2021-009) to M.P. and L.B.; the HI-TRON strategy project PACESSETTING and the DKTK Joint Funding Program INNOVATION INVENT4GB to W.W., M.P., and L.B. The DFG, project 404521405 (SFB1389 UNITE Glioblastoma B01) to T.B. and M.P. Y-C.C. received an international fellowship from the Helmholtz International Graduate School at DKFZ and was supported by RTG 2099 and UNITE SFB1389. A.C.D. was supported by Mildred-Scheel scholarship. P.K. and D.A.A. were funded by RTG 2099 and UNITE SFB1389. K.A. was supported by Heinrich F.C. Behr-Stipend.

## Author contributions

Y-C.C. conceptualized the study, designed and performed experiments, analyzed and interpreted data, and wrote the manuscript. A.C.D., D.A.A., A.D.R., A.K., M.K., and C.K. performed in vitro and in vivo experiments and edited the manuscript. X.M. and H-K.L. established and processed organoids and edited the manuscript. P.K. and B.Z. analyzed patient and

organoid scRNA-seq data and edited the manuscript. A.K.S. and F.S. collected glioblastoma histology samples and edited the manuscript. M.S., H.B., T.B., R.H., and W.W. interpreted data and edited the manuscript. A.B. and V.M.P. predicted off-targets. K.A., J.P.B., and A.B.R. performed the ligandomics analysis and edited the manuscript. H.F., M.R., and N.E. collected patient tumor samples, established primary GB cell lines and edited the manuscript. M.P. supervised the study, interpreted the data, provided funding and edited the manuscript. E.W.G. provided cloning methodology, retrieved the TCR and edited the manuscript. L.B. conceptualized and supervised the study, interpreted the data, provided funding and wrote the manuscript.

## Funding

## Competing interests
The identified reactive TCR was patented by Y-C.C., M.K., C.K., E.W.G., W.W., M.P. and L.B., "T cell receptor derived binding polypeptides" (WO 2023/213904). E.W.G. and M.P. are founders of TCellTech. The remaining authors declare no competing interests.

## Additional information

[1]Clinical Cooperation Unit (CCU) Neuroimmunology and Brain Tumor Immunology, German Cancer Research Center (DKFZ), Heidelberg, Germany. [2]German Cancer Consortium (DKTK), DKFZ, core center Heidelberg, Heidelberg, Germany. [3]Faculty of Biosciences, Heidelberg University, Heidelberg, Germany. [4]Department of Neurology, Medical Faculty Mannheim, Mannheim Center for Translation Neuroscience (MCTN), Heidelberg University, Mannheim, Germany. [5]Division of Molecular Neurogenetics, DKFZ, DKFZ-ZMBH alliance, Heidelberg, Germany. [6]DNA Vector Laboratory, DKFZ, Heidelberg, Germany. [7]Neurology Clinic, Heidelberg University Hospital, Heidelberg, Germany. [8]CCU Neurooncology, DKFZ, Heidelberg, Germany. [9]Ann Romney Center for Neurologic Diseases, Brigham and Women's Hospital, Harvard Medical School, Boston, MA, USA. [10]Institute for Pathology, Department of Neuropathology, Heidelberg University, Heidelberg, Germany. [11]CCU Neuropathology, DKFZ, Heidelberg, Germany. [12]Institute for Clinical Transfusion Medicine and Cell Therapy, Heidelberg, Germany. [13]Institute for Immunology, Heidelberg University Hospital, Heidelberg, Germany. [14]Faculty of Medicine, Goethe University, Frankfurt a.M., Frankfurt, Germany. [15]Institute for Transfusion Medicine and Immunohematology, German Red Cross Blood Service Baden-Württemberg-Hessen, Frankfurt a.M., Frankfurt, Germany. [16]Ardigen, ul. Podole 76, Kraków, Poland. [17]Division of Immunotherapy and Immunoprevention, DKFZ, Heidelberg, Germany. [18]Molecular Vaccine Design, German Center for Infection Research (DZIF), partner site Heidelberg, Heidelberg, Germany. [19]Department of Neurosurgery, University Hospital Mannheim, Mannheim, Germany. [20]Immune Monitoring Unit, National Center for Tumor Diseases (NCT), NCT Heidelberg, a partnership between DKFZ and Heidelberg University Hospital, Heidelberg, Germany. [21]Helmholtz Institute for Translational Oncology Mainz (HI-TRON Mainz) – A Helmholtz Institute of the DKFZ, Mainz, Germany. [22]DKFZ Hector Cancer Institute at the University Medical Center Mannheim, Mannheim, Germany. ✉e-mail: l.bunse@dkfz.de

