## [Transparent Peer Review file · Nature Communications]

Vaccine-induced T cell receptor T cell therapy targeting a glioblastoma stemness antigen

Corresponding Author: Dr Lukas Bunse

Version 0:

Reviewer comments:

Reviewer #1

(Remarks to the Author)

I got the opportunity to review this fascinating manuscript. In the present study, the authors present a novel therapeutic option based on PTPRZ1-TCR-T. This strategy, for which a phase 1 study is in preparation, allows for avoiding tonic signaling and targeting a glioblastoma cell stemness antigen.

Some points need to be addressed.

1. The first part of the study was performed using the U87 TMG cell line. I understand the reason for choosing U87, considering they are a well-characterized HLA-A02 cell line. However, U87 MG are not completely representative of a glioblastoma, for example, they are methylated in MGMT promoter. Therefore, did the authors compare the STR profile with their U87 cells to those of U87MG from ATCC?

2. I need to better understand why the authors observed “no therapeutic effect” using an IV administration as a single therapeutic modality, but a preclinical response when combined with IV and ICV.

3. Based on the evidence that pleiotrophin (PTN) is secreted by TAM, it is crucial to consider the potential influence of the tumor microenvironment on the efficacy of this therapeutic strategy. Did the authors consider a combination treatment? Chemotherapy or radiotherapy, even for modulating the tumor microenvironment?

The methodology is complete. The abstract is very clear and the manuscript is very well written.

Reviewer #2

(Remarks to the Author)

This is a very well done study, with rigorous experimental controls and datasets.

I do think that there are some potential limitations of the product that could be addressed in the Discussion. These include potential pitfalls such as:

(1) Murinization of the TCR constant regions may be immunogenic

(2) It is not clear how much antigen needs to be presented to effect clinical responses (the U87-transduced with the tandem mini-gene is great, but obviously has high antigen presentation); the spheroid data is helpful but is limited by in vitro nature.

(3) It is not clear if targeting stemness (GSC's) will be sufficient for clinical responses in GBM.

However, these are all things that will become clearer in the clinical trial – my only suggestion is that they be discussed, I don't think it's necessary to address these experimentally in this paper.

Additional minor points:

1. Figures should be numbered/labeled.

2. Figure 3H is confusing with the red PTPRZ TCR and the red CD8 and the blue Flu TCR and CD4. Figure 3K has a lot of colors for unclear reasons – donor, pink, and biorender cells. This needs to be simplified and clarified with simple labels instead of drawings.

3. In Figure 4, why do the authors think that there is no anti-tumor response after the first TCR injection (between day 9 and

day 16?) is it because the tumor are too small to measure in that time? How accuratelyl can < 20 mm2 be measured?

Version 1:

Reviewer comments:

Reviewer #1

(Remarks to the Author)

The authors have adequately responded to all the comments.

I confirm that this work represents an exceptional study suitable for publication in Nature Communications.

S.Pellegatta

Reviewer #2

(Remarks to the Author)

the authors have addressed my concerns

Transgenic T cell therapy targeting the glioblastoma stem cell antigen PTPRZ1 with a vaccine-induced, patient-derived T cell receptor (NCOMMS-24-46223-T)

Reviewer Comments

Point-by-point responses in red

Reviewer #1:

I got the opportunity to review this fascinating manuscript. In the present study, the authors present a novel therapeutic option based on PTPRZ1-TCR-T. This strategy, for which a phase 1 study is in preparation, allows for avoiding tonic signaling and targeting a glioblastoma cell stemness antigen.

We thank the referee for this highly positive evaluation of our work.

Some points need to be addressed.

1. The first part of the study was performed using the U87 TMG cell line. I understand the reason for choosing U87, considering they are a well-characterized HLA-A02 cell line. However, U87 MG are not completely representative of a glioblastoma, for example, they are methylated in MGMT promoter. Therefore, did the authors compare the STR profile with their U87 cells to those of U87MG from ATCC?

Our U87 line is U87 MG purchased from ATCC first used in our group in Weiler et al., 2014. Materials and methods are now updated. We agree with the referee that U87 endogenously expressing HLA-A02 is not completely representative of glioblastoma. We therefore established and included several other HLA-A02-positive and HLA-A02-negative primary glioblastoma cell lines and showed HLA-A02-specific killing (Figure 5E).

2. I need to better understand why the authors observed “no therapeutic effect” using an IV administration as a single therapeutic modality, but a preclinical response when combined with IV and ICV.

We thank the referee for this interesting question currently intensively discussed in the field. In a preclinical brain tumor CAR-T study led by others, providing first evidence of differential therapeutic efficacy dependent on the route of cell transfer, intravenous administration revealed diminished efficacy in comparison to intracerebroventricular transfer (Priceman et al., *Clin Cancer Res* 2018). Moreover, there is increasing evidence that this is also the case in clinical glioblastoma patients (Bagley et al., *Nat Cancer* 2024). Conversely, responses to CD19-CAR-T in the context of primary/secondary CNS lymphoma have been reported, suggesting that limited efficacy of intravenous therapy is not exclusively driven by tumor outgrowth “behind” the blood brain barrier (Frigault et al., *Blood* 2022). Therefore, we did not disregard a potential but marginal therapeutic efficacy of intravenous cell transfer and therefore combined it with intracerebroventricular treatment in our experimental GB model, resulting in potent tumor control (Figure 4C-F). The same treatment regime is currently planned for the first-in-human clinical trial (<https://doi.org/10.1093/neuonc/noad137.145>).

3. Based on the evidence that pleiotrophin (PTN) is secreted by TAM, it is crucial to consider the potential influence of the tumor microenvironment on the efficacy of this therapeutic strategy.

Did the authors consider a combination treatment? Chemotherapy or radiotherapy, even for modulating the tumor microenvironment?

We thank the referee for this comment and agree that combinatorial treatments against GB to tackle the immunosuppressive microenvironment in addition need to be developed. Indeed, it has been reported that M2-like macrophages produce PTN driving GSC-tumor outgrowth (Shi, *Nat Commun*, 2017). We therefore concluded for our study that immune-targeting of PTPRZ1 could eradicate a highly relevant tumor cell population even in heterogenous tumors. It would be tempting to speculate that targeting PTN within the tumor microenvironment would lead to further upregulation of its target PTPRZ1, hence, making GSC even more susceptible for PTPRZ1-specific TCR-T. As this hypothesis would require investigation in an immunocompetent MHC-humanized mouse glioblastoma model (HLA-A02-positive) following combined TCR-T and PTN blockade, we conclude that this could be investigated indeed in a follow-up study. In regard to the first-in-human clinical trial, we will have to assess the safety and efficacy in recurrent glioblastoma first due to regulatory recommendations. Here, in the majority of cases, patients do not receive re-irradiation. Of note, overall, we definitively agree with the referee and have combined T-cell driven immunotherapies (vaccines) in combination with SOC irradiation in the past (Platten, *Nature*, 2021; Graßl, *Nat Med*, 2023).

The methodology is complete. The abstract is very clear and the manuscript is very well written.

We again thank the referee for this very positive feedback.

Reviewer #2:

This is a very well done study, with rigorous experimental controls and datasets. I do think that there are some potential limitations of the product that could be addressed in the Discussion. These include potential pitfalls such as:

(1) Murinization of the TCR constant regions may be immunogenic

We thank Reviewer 2 for valuing our work and the thorough review of our manuscript. In previous discussions with our national regulatory authorities, we had discussed the immunogenicity of murine constant TCR regions, too. Overall, murinization of constant TCR alpha/beta chains to reduce mispairing with endogenous patient-individual TCRs is well-investigated, improves avidity, and confers high tumor reactivity (NCT03190941; Wang, *Cancer Immunol Res*, 2014). Although the murine TCR constant regions are possibly immunogenic, at the same time, their use inhibits the generation of a potentially auto-reactive *de novo* TCR that is generated when transgenic TCRs pair incorrectly with patient individual alpha and beta TCR chains. In previous clinical studies using murine TCR-engineered T cells, some patients exclusively developed antibodies directed to the murine TCR variable regions but not to the constant regions common to all murine TCR. Hence, patients treated with murine TCR variable regions can develop an immune response to gene-modified cells in a minority of cases, but this did not affect clinical outcome (Davis et al., *Clin Cancer Res* 2010). We have now included a sentence on this topic in the discussion. The biological consequences

of TCR murinization will be carefully assessed in the objectives of our upcoming clinical trial by using anti-mTRBC ELISA and ELISpot assays.

(2) It is not clear how much antigen needs to be presented to effect clinical responses (the U87-transduced with the tandem mini-gene is great, but obviously has high antigen presentation); the spheroid data is helpful but is limited by in vitro nature.

We thank the referee for this important comment. We agree that for assessing the required antigen level for therapeutic efficacy, U87 TMG line is suboptimal. At the same time, with our in vivo model, we primarily aimed to demonstrate sufficient intratumoral infiltration and effector functions of TCR-T cells following intracerebroventricular injection. To approach the question of PTPRZ1 antigen levels required for TCR-T efficacy, we now provide additional experimental data. We assessed the expression levels of PTPRZ1 in our panel of generated primary spheroid GB cell lines (SFigure 10E). Importantly, in comparison to MA140, which demonstrated an approximately 4-fold PTPRZ1 compared to the well-studied cell line P3, MA120 showed only moderate PTPRZ1 upregulation normalized to P3 (SFigure 10E), but T cell killing was comparable (Figure 5E). This finding suggests that PTPRZ1-TCR-T is able to lyse cells across different expression levels. To prove target-dependent killing even at relatively low PTPRZ1 expression levels, we generated oligoclonal *PTPRZ1* KO lines from D170_44, displaying a moderate PTPRZ1 overexpression (SFigure 10E), with two different guide RNAs (Figure 5I). In these engineered primary GB cells, we now show that a reduction of mean PTPRZ1 expression of approx. 60% abolishes TCR-T cytotoxicity (Figure 5J). Beyond this demonstration of target-dependent killing even at relatively low expression levels of PTPRZ1 compared to other GB cell lines, we plan to correlate *PTPRZ1* expression levels and outcome in exploratory post-hoc analyses in the first-in-human clinical trial.

(3) It is not clear if targeting stemness (GSC's) will be sufficient for clinical responses in GBM. However, these are all things that will become clearer in the clinical trial – my only suggestion is that they be discussed, I don't think it's necessary to address these experimentally in this paper.

With the longitudinal interrogation, our TCR-T product shows a preference for targeting GSCs yet still effectively lyses differentiated fast-cycling cells (Figure 5B&C). Overall, we agree with the referee that demonstrating objective responses will be the most important outcome of the clinical trial. The limitation and safety concerns are now further discussed in the manuscript.

Additional minor points:

1. Figures should be numbered/labeled.

We thank for this comment and now label the figures.

2. Figure 3H is confusing with the red PTPRZ TCR and the red CD8 and the blue Flu TCR and CD4. Figure 3K has a lot of colors for unclear reasons – donor, pink, and biorender cells. This needs to be simplified and clarified with simple labels instead of drawings.

We thank for the comment and now improve the readability of these figures.

3. In Figure 4, why do the authors think that there is no anti-tumor response after the first TCR injection (between day 9 and day 16?) is it because the tumor are too small to measure in that time? How accurately can < 20 mm² be measured?

Thanks for pointing out this interesting observation. We hypothesize that this phenomenon is possibly linked to inflammation-associated edema, hence, tumors do not shrink immediately (Mahdi, Nat. Med., 2023). We think that down to 10 mm², tumors are palpable and can be accurately measured with the calipers we apply.